# Ultrafast-Laser Micro-Structuring of LiNi_0.8_Mn_0.1_Co_0.1_O_2_ Cathode for High-Rate Capability of Three-Dimensional Li-ion Batteries

**DOI:** 10.3390/nano12213897

**Published:** 2022-11-04

**Authors:** Minh Xuan Tran, Peter Smyrek, Jihun Park, Wilhelm Pfleging, Joong Kee Lee

**Affiliations:** 1Division of Energy and Environmental Technology, KIST School, Korea University of Science and Technology (UST), Deajeon 34113, Korea; 2Center for Energy Storage Research, Green City Research Institute, Korea Institute of Science and Technology (KIST), Seoul 02792, Korea; 3Karlsruhe Institute of Technology, Institute for Applied Materials, P.O. Box 3640, 76021 Karlsruhe, Germany; 4Karlsruhe Nano Micro Facility, Hermann-von-Helmholtz-Platz 1, 76344 Eggenstein-Leopoldshafen, Germany; 5APC Technology, 108 68 Gangbyeonyeok-ro-4-gil, Gwangjin-gu, Seoul 05116, Korea

**Keywords:** three-dimensional batteries, LiNi_0.8_Mn_0.1_Co_0.1_O_2_ cathode, femtosecond ultrafast laser, electrode micro-structuring

## Abstract

Femtosecond ultrafast-laser micro-patterning was employed to prepare a three-dimensional (3D) structure for the tape-casting Ni-rich LiNi_0.8_Mn_0.1_Co_0.1_O_2_ (NMC811) cathode. The influences of laser structuring on the electrochemical performance of NMC811 were investigated. The 3D-NMC811 cathode retained capacities of 77.8% at 2 C of initial capacity at 0.1 C, which was thrice that of 2D-NMC811 with an initial capacity of 27.8%. Cyclic voltammetry (CV) and impedance spectroscopy demonstrated that the 3D electrode improved the Li^+^ ion transportation at the electrode–electrolyte interface, resulting in a higher rate capability. The diffusivity coefficient *D*_Li+_, calculated by both CV and electrochemical impedance spectroscopy, revealed that 3D-NMC811 delivered faster Li^+^ ion transportation with higher *D*_Li+_ than that of 2D-NMC811. The laser ablation of the active material also led to a lower charge–transfer resistance, which represented lower polarization and improved Li^+^ ion diffusivity.

## 1. Introduction

The conventional Li-ion batteries (LIBs) have several limitations, such as low energy density, high cost of the active material, and poor thermal stability [1,2,3]; hence, their use is limited. However, LiNi_x_Mn_y_Co_z_O_2_ (NMC) has been attracting increasing attention as a promising cathode material with a higher specific capacity and energy density than that of the conventional LiCoO_2_ (LCO) [4,5]. The Ni-rich NMC cathode, particularly LiNi_0.8_Mn_0.1_Co_0.8_O_2_ (NMC811), is more cost efficient and has a higher practical specific capacity (200 mAh g^−1^) than that of the lower-Ni-contained NMC cathode (160 mAh g^−1^) at an average discharge potential of 3.8 V (vs. Li^+^/Li) [6,7]. However, batteries for electric vehicles and applications in smart grid systems require high power to sufficiently charge in a very short time. The output power of LIBs is determined by the transport rate of the Li^+^ ion in an electrolyte and active material. Although the NMC cathode materials exhibit superior features in comparison with those of LCO, the NMC cathode is unable to demonstrate the high rate capabilities of conventional electrodes caused by structure disruption and cation-mixing between Li^+^ and Ni^2+^ [8,9,10]. Moreover, the final cell weight and specific cell capacity are functions of the cathode film thickness [11]. Thus, when the thickness of the cathode film increases from 50 µm to 200 µm, the specific cell capacity increases by 29% [12]. Electrodes with thicknesses of 100–200 µm exhibit high energy density, whereas thin electrodes with thicknesses of 10–50 µm exhibit high power density [13,14]. However, the Li^+^ diffusion kinetics and mechanical integrity of the film decrease with the electrode film thickness because the active material expands and contracts during continuous charge/discharge processes [15]. Three-dimensional (3D) electrode configuration in LIBs is one of the approaches to overcome the limitations of thick electrode films, such as power losses and mechanical degradation during charge and discharge operations because of high volume expansion [16,17]. Three-dimensional structuring of electrodes improves the electron and ion diffusion kinetics in the electrodes. Moreover, 3D-electrode architectures lead to an increased active surface area, reduced mechanical tensions during electrochemical cycling, and an overall reduced cell impedance [18]. Common methods for 3D configuration in electrodes are the structuring of the substrate or current collector via template deposition of nano-rods or anisotropic etching of silicon. These approaches are only available for thin-film micro batteries and are not feasible for thick-film electrodes [19]. Nonetheless, ultrafast laser structuring is a novel approach to realize the 3D battery concept, and has been applied for thin-film electrodes, including LiCoO_2_, Li_2_Mn_2_O_4_ and SnO_2_ [17,20,21]. Moreover, composite electrodes that were patterned with ultrafast lasers have exhibited improved capacities at high charge and discharge currents. The enhancement in the electrochemical performance can also be attributed to the improvement in the wetting of the electrode surface, which results in a decrease in the time and cost required for the vacuum and storage processes during the assembly of LIB cells [12].

In this work, commercial NMC811 was employed as the active material to obtain high energy density. Femtosecond ultrafast laser structuring technology was applied to tape-casted composite electrodes to improve the rate capability of LIBs. In order to investigate the balance of active material loss by ablation from laser structuring and increase in interfacial surface area, ultrafast laser patterning was carried out on NMC811 cathodes with two different thicknesses of 40 and 100 µm. Without structuring, the thin film electrode could deliver better rate capability. When 3D structuring was performed, the rate capability of the thick film electrode was improved to 133 mAh g^−1^ at 5 C. Meanwhile, the structured thin film electrode could not deliver capacity as high as that of post-structuring due to more severe loss of active material. Therefore, the balancing of loss material and increment of interfacial area was a factor that affected improvement in rate capability. The diffusivity of Li^+^ ions was studied using cyclic voltammetry (CV) and electrochemical impedance spectroscopy. When laser structuring of the electrode was performed, the transportation of the Li^+^ ions improved significantly, with a higher value of D_Li+_, which could be attributed to the high aspect ratio of 3.7 and increase in surface area of 78%.

## 2. Materials and Methods

### 2.1. Material Characterization

The commercial NMC811 powders, with particle size of 10 µm (ECOPro BM, Chungcheongbuk-do, Korea), were used as obtained as the electrode active material. X-ray diffraction (XRD, Rigaku, Tokyo, Japan) analysis was performed using Cu *K*α radiation (*λ* = 0.15406 Å) in the 2*θ* range of 20–80° at a continuous scan mode with a step size of 0.02° and a scan rate of 2° min^−1^. The morphology and structure of the NMC811 particles were observed using field-emission scanning electron microscopy (FE-SEM, Inspect F50, ThermoFisher, Hillsboro, OR, USA) with an accelerating voltage of 10 kV.

### 2.2. Ultrafast-Laser Structuring of NMC811 Electrode 

The cathode slurry, comprised of NMC811: acetylene black (DB100, Denka, Tokyo, Japan): polyvinylidene fluoride (PVDF, Aldrich, MO, USA), in a weight ratio of 85:10:5, was well mixed in *N*-methylpyrrolidone (NMP, Aldrich, MO, USA) solvent. Then, the slurry was tape-casted on an 18 µm thick aluminum foil as the current collector. After being dried in an oven at 60 °C overnight, a calendaring roll pressure was applied to reduce 10% of the original thickness of the as-prepared cathode to decrease the porosity of the electrodes. The average thickness of the cathodes was 100 µm (exclusive of the Al foil thickness).

Ultrafast-laser-assisted structuring was performed on the calendared electrodes using a fiber laser (Tangerine, Amplitude Systèmes, Pessac, France) operating at a wavelength of λ = 1030 nm with a pulse duration of 380 fs and a laser pulse repetition of 500 kHz. The laser beam was scanned over the sample surface using deflection mirrors with scanning velocities in the range of 100–1500 mm s^−1^. All experiments were conducted under ambient air and the ablated material was removed by an exhaust.

### 2.3. Electrochemical Tests

The 2032 coin cell was prepared after the electrodes were dried in a vacuum oven at 80 °C for 12 h. The cathodes were laser cut into discs of 12 mm diameter. A Li metal foil (Wellcos Co., Gyeonggi-do, Korea) disk of *Ø*18 mm was used as the anode. The cathode and anode in the coin cells were separated by microporous polypropylene (PP) membranes (Celgard 2400, North Carolina, USA) in an electrolyte of 1 M LiPF_6_ in ethylene carbonate (EC): ethyl methylene carbonate (EMC): diethylene carbonate (DMC) (1:1:1 volume ratio). The cell assembly was performed in a dry room with a dew point temperature of less than −100.2 °C. The electrochemical testing experiments were conducted using a Maccor automated battery tester (MACCOR series-4000, Tulsa, OK, United States) at room temperature of 25 °C. The Li-ion cells were galvanostatically charged/discharged at various currents in the working voltage range of 3.0–4.3 V, after being allowed to rest for 12 h. CV was conducted at scan rates of 0.3–1 mV s^−1^ in the potential range of 3.0–4.3 V. The coin cell was measured by electrochemical impedance spectroscopy (EIS) by applying a frequency range of 1 mHz to 100 kHz to a potentiostat (Bio-Logic Science Instruments, Seyssinet-Pariset, France) with a voltage amplitude of 5 mV.

### 2.4. Diffusivity Coefficient Calculation

#### 2.4.1. Diffusivity Coefficient by Cyclic Voltammetry

The diffusion coefficient of lithium ion can be determined by the Randles–Sevcik equation:(1)$$i_{P}=0.4463\;\mathrm{mAF}{\left(\frac{F}{RT}\right)}^{1/2}C_{\mathrm{Li}}D_{\mathrm{Li}}^{1/2}\omega \vartheta^{1/2},$$
where *i_P_* is peak current (A), *m* is the mass of active materials-NMC811 (g), *A* is the effective area of the electrode (cm^2^), *F* is the Faraday constant (96,485 s A mol^−1^), *R* is the gas constant (8.314 J K^−1^ mol^−1^), T is the absolute temperature (K), *C* is the initial concentration of lithium ion (1.0 mol cm^−3^), *D*_Li+_ is the chemical diffusion coefficient of Li^+^ (cm^2^ s^−1^), and *ν* is the scan rate in mV s^−1^ [22].

#### 2.4.2. Diffusivity Coefficient by EIS

The diffusivity coefficient is calculated using the following equation:(2)$$D_{{\mathrm{Li}}^{+}}=\frac{R^{2}T^{2}}{2A^{2}n^{4}F^{4}C^{2}\sigma^{2}},$$
where *A* (cm^2^) is the effective surface area of the electrode, *n* is the number of exchanged electrons (*n* = 1), *F* is Faraday constant, *R* is gas constant, *T* is temperature (285 K), *C* is the initial concentration of Li^+^ in electrolyte, and *σ* is the Warburg factor [23].

The Warburg factor can be determined with the real resistance according to the equation below:(3)$$Z_{re}=R_{s}+R_{ct}+\sigma \omega^{1/2},$$
where *R_s_* (Ω) denotes solution resistance, and *ꞷ* (Hz) is frequency. The value of σ is extracted from the slope of linear correlation between *Z_re_* and *ꞷ^−^*^1/2^.

The effective surface area of electrode (*A*) used in both Equations (1) and (2) was estimated as the surface area of the electrode. For the 2D-NMC811 electrode, the value of *A* was calculated as the surface area of the electrode with a diameter of *Ø*1.2 cm. Effective surface area of 3D-NMC811 was calculated based on the increasing surface area, as summarized in Appendix A.

## 3. Results and Discussion

The as-received NMC811 particles had a uniform spherical shape with average diameters, D50, of approximately 10 µm, as depicted by the FE–SEM images in Appendix A. The NMC811 spheres were primarily comprised of polyhedral grains of sub-micron size that aggregated to form secondary NMC811 particles, which were consistent with other reports [24,25]. As shown in Appendix A, the result from the EDX of the selected area suggested that the surface of the NMC811 particles was represented by a composition of a Ni:Mn:Co atomic ratio of 8:1:1, approximately. The crystalline structure of NMC was confirmed by conducting XRD measurements. As illustrated in Appendix A, the XRD patterns indicated that NMC811 had a layer structure, based on hexagonal *α*-NaFeO_2_ with space group R3 m. There were no impurity or secondary phases detected. NMC811 powder had a highly well-defined layer structure, indicated by the appearance of peak splitting of 006/102 and 108/110 (Appendix A) [26,27]. Moreover, the intensity ratio of 003 and 104 peaks, *I*_(003)_/*I*_(104)_, was 1.1, approximately, which indicated a low degree of cation mixing and a good layered structure [7].

The morphologies of the electrodes before and after the structuring by the ultrafast laser are illustrated in Figure 1. After calendaring of the electrodes using a rolling press, the thicknesses of the electrodes were determined as 40 µm and 100 µm, approximately, indicated by the microscopy images in Figure 1. Both electrodes denoted as 2D-NMC811 were used for the laser structuring experiments. The femtosecond laser could remove the composite active materials down to the Al current collector. As depicted in Figure 1a,b, the electrodes were patterned to linear structure with a pitch distance of 200 µm. Figure 1c,d illustrates cross-sectional images of the structured electrodes, denoted as 3D-NMC811. The laser ablated active materials deep to the Al current collector to form a V-shaped channel. The 40 µm-thick NMC811 channel dimensions were determined to be ~25 μm wide and ~38 μm deep, while the 100 µm-thick cathode channels were found to be ~25 μm wide and ~90 μm deep. The edges of the cathode channels were observed to be smooth and uniform. 

In order to compare with ablation studies on electrodes with different thicknesses, the aspect ratio *AR* (channel depth divided by channel width of half height), amounts of material loss and interfacial area increments were estimated and listed in Appendix A. When the thickness of the electrodes increased from 40 µm to 100 µm, the *AR* value and interfacial area increment increased 4 times, from 0.96 to 3.72, and 20% to 78%, respectively. Along with the *AR* value, loss of active material by ablation from laser structuring decreased from 10% to 6.4% for the 40 µm- and the 100 µm-thick electrodes, respectively. The changes in the above parameters suggested that the laser structuring of electrodes would have more significant effect to improve electrochemical performance of the thick film, 3D-NMC811.

Figure 2a,b exhibits the rate capabilities of 2D- and 3D-NMC811 at various C-rates with thicknesses of 40 and 100 µm, respectively. The 40-µm-thick 2D-NMC811 cathode showed high capacities of 165, 102 and 66 mAh g^−1^ at C-rates of 0.1 C, 1 C and 2 C, respectively. When the thickness of the cathode increased to 100 µm, the capacities at 2 C and 5 C decreased drastically to 48 and 6 mA h g^−1^, respectively, which was 2 times lower than that of the 40-µm cathode. The deterioration of capacity at high current density in the thick electrode was caused by two aspects: (1) the longer diffusion and migration path of Li^+^ ions and (2) the higher local ion-current densities at the electrode/separator interface [28,29]. However, when the ultrafast-laser-structuring was performed, the tendency of rate capability changed remarkably. The 40-µm-thick cathode showed worsened performance with capacity at a high C-rate of 2 C and 5 C decreasing twice to 76 and 31 mAh g^−1^, respectively, compared to that of 2D-NMC811. On the other hand, the thick 3D-NMC811 electrode (100 µm) showed capacities at 2 C increasing thrice from 48 to 133 mAh g^−1^, which was higher than that of thin 3D-NMC811 at the same C-rate, as shown in Appendix A. As mentioned above, 3D laser-structuring can drastically improve the electrolyte wetting, and the active material ablation from laser patterning could provide artificial porosity and an electrolyte reservoir. Thus, laser-induced micro-structuring of the electrode could enhance the electrochemical activation of NMC811 active particles in deeper layers, which resulted in an improvement in the rate capability of thick-film electrodes. Moreover, although interfacial surface area and porosity could increase by means of laser-induced ablation, the negative effect observed for the thin film electrode could be attributed to loss of up to 10% of active material and low *AR* value.

To study the promotion of Li^+^ transportation in the case of a 100-µm-thick structured electrode, CV was performed. The CV curves of NMC811 at a scan rate of 0.3 mV s^−1^ (Figure 3a,b) indicated three distinct anodic/cathodic peaks, which could be assigned to distinct features of the Ni-rich NMC cathode materials. In the positive sweep, the first anodic peaks in the region of 3.4–3.8 V were assigned to the phase transition from a hexagonal to a monoclinic (H_1_→M) lattice. In the region of potential higher than 3.8 V, there were two anodic peaks at approximately 4.0 and 4.2 V, which were caused by the phase transition from M→H_2_ and H_2_→H_3_, respectively [30,31]. In the first anodic scan, the potential downshifting could be attributed to the initial activation and stabilization of the active material. The current in the first anodic peak did not decrease drastically for the subsequent sweep scan in 3D-NMC811. In contrast, there was a significant change in the intensity of current from the first scan to the following scans in 2D-NMC811. This result indicated a decrease in the interfacial polarization and better reversibility of the 3D-electrode.

The potential of the cells with 2D- and 3D-NMC811 were swept at scan rates of 0.03–1.0 mV s^−1^ to calculate the diffusion coefficient (*D*_Li+_). As illustrated in Figure 3c,d, the value of the peak currents increased proportionally with the increase in the scan rates, because of the increase in the flux of the charge carrier species at the electrode surface [32]. Since the polarization increased with scan rates, the redox peak positions also shifted to higher and lower voltages for the oxidation (anodic) and the reduction (cathodic) processes, respectively. The redox peak potentials of the structured electrode were significantly higher than those of the unstructured electrode at the same scan rate. The differences related to the polarization influence were more obvious at a sweep rate higher than 0.5 mV s^−1^. The mass transfer at the electrode–electrolyte interface was a major factor that caused the polarization in Li-ion cells [33]. The redox peak positions in the CVs upshift and downshift in the charge and discharge processes, respectively, indicated that the enhancement in the mass transfer led to the hindering of the polarization. The 3D-structuring of the electrode by an ultrafast laser could improve the rate performance of NMC811 by enhancement of mass transfer upon electrode. In addition, the diffusion coefficient of the Li^+^ ions could be determined by Equation (1) [34]. As depicted in Appendix A, a linear correlation of the peak current density (*j*) and the square root of scan rate (*ν*^1/2^) was obtained. Accordingly, the diffusivity coefficients of 2D-NMC cathode were 16.6 × 10^−11^ and 2.0 × 10^−11^ cm^2^ s^−1^ for the oxidation and reduction processes, respectively, as shown in Appendix A. When laser patterning was employed, there was an increase in *D*_Li+_ for the anodic peak to 18.9 × 10^−11^ cm^2^ s^−1^. This result implied that laser structuring of the electrode could improve the Li^+^ ion diffusivity in the 3D-NMC811 electrode, due to the increase in the interfacial surface area and improvement in the electrolyte wetting, thereby facilitating fast Li^+^ ion transportation.

Moreover, to understand more about the effect of laser structuring on the electrochemical characteristics, impedance analysis was conducted, as illustrated in Figure 4a. The impedance spectroscopy results were consistent with results from CV and diffusivity coefficient. A semicircle in the high frequency range, representing charge transfer resistance (*R*_ct_), and a sloping line in the low frequency range, representing Warburg resistance, were observed in both the 2D- and 3D-electrodes. In the Nyquist plots, the semicircle of the 3D-electrode was smaller than that of the 2D electrode. In general, the semicircle in the Nyquist plot represented the impedance values related to the charge–transfer resistance (*R*_ct_) obtained by fitting the Nyquist plots, using ZView software, with the corresponding equivalent circuit (Appendix A). Moreover, the diffusivity coefficient could also be calculated using *σ* from Equation (2), which was the Warburg factor extracted from the slope of linear correlation between *Z*_re_ and *ꞷ*^−1/2^, as depicted in Appendix A [23]. As shown in Figure 4b and Appendix A, fresh cells with 3D-NMC811 possessed a lower *R*_ct_ of 107.7 Ω and a higher *D*_Li+_ of 3.5 × 10^−11^ cm^2^ s^−1^ than those of the 2D-NMC811 electrode (136.6 Ω and 1.6 × 10^−11^ cm^2^ s^−1^). The above results indicated that the 3D-patterning of electrodes could improve Li^+^ transportation and decrease electrode polarization [23,35]. The differences in *R*_ct_ and *D*_Li+_ values between 2D- and 3D-NMC811 were well-correlated with the data obtained from the CV studies. This result demonstrated that the 3D-structuring of the composite cathode could improve the transportation of Li ions, resulting in enhancement of the rate capability.

## 4. Conclusions

Commercially available Ni-rich LiNi_0.8_Mn_0.1_Co_0.1_O_2_ (NMC811), with a high capacity and energy density, was used as the cathode material for a Li ion battery. The tape-casted composite electrodes were micro-structurally modified by a femtosecond ultrafast laser to generate 3D-electrode configuration. The 3D-patterning of electrodes with different thicknesses resulted in different aspect ratio values and material losses. Higher aspect ratio and increment of interfacial surface area, and lower active material loss were observed in the thick-film 3D-NMC811. The laser structuring improved the capacity of the thick film cathode at a high rate but failed to have a positive effect on that of the thin film electrode. The 3D-modification of the cathode improved the mass transfer of Li^+^ ions into the deep layers of the electrode, as well as the wetting capability of the electrolyte. The diffusivity coefficient of the Li^+^ ions, determined by CV and EIS, demonstrated that the 3D-NMC811 possessed a lower value of *D*_Li+_ than that of 2D-NMC811. The charge transfer resistance was lower, which indicated a lower polarization and improved Li^+^ ion diffusivity. Faster ion transportation in 3D-NMC811 resulted in a higher rate capability of 133 mAh g^−1^ at a C-rate of 2 C, which was 77.8% of the initial capacity at 0.1 C and thrice that of 2D-NMC811. Laser structuring to realize the 3D-battery concept could be an approach to commercialize high-energy NMC811 cathode materials with high power output.

## Figures and Tables

**Figure 1 nanomaterials-12-03897-f001:**
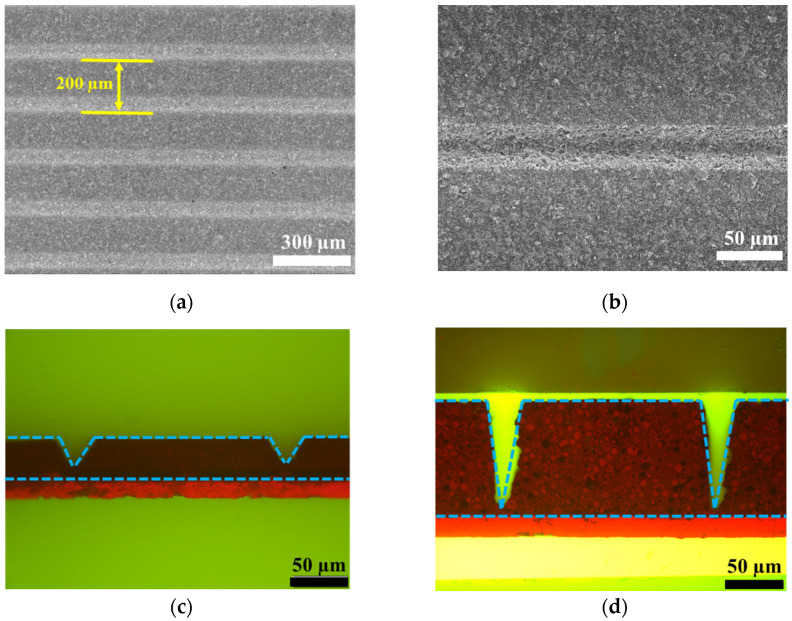
SEM and microscopic images of NMC electrodes (**a**,**b**) top-view of structured electrodes (SEM) at different magnifications, cross section of electrode at (**c**) 40 µm and (**d**) 100 µm (microscope).

**Figure 2 nanomaterials-12-03897-f002:**
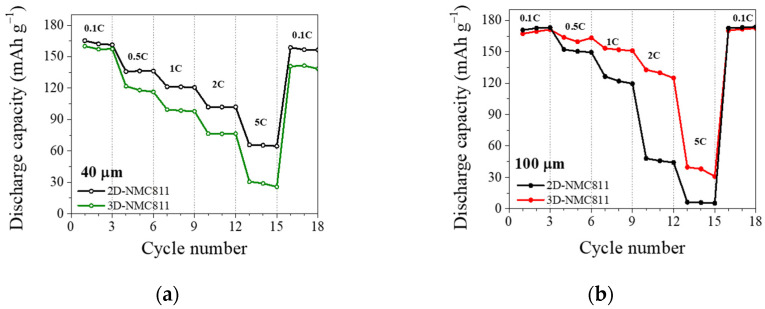
Rate capability performance for unstructured and structured NMC electrodes with thicknesses of (**a**) 40 µm and (**b**) 100 µm.

**Figure 3 nanomaterials-12-03897-f003:**
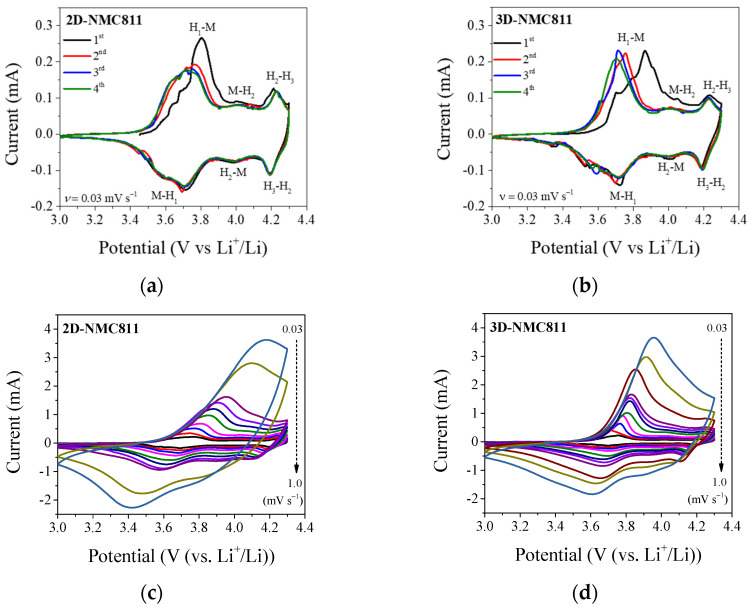
Cyclic voltammetry (CV) data of 2D- and 3D-NMC811 electrodes at (**a**,**b**) 0.3 mV s^−1^ and (**c**,**d**) various scan rates from 0.03 to 1.0 mV s^−1^.

**Figure 4 nanomaterials-12-03897-f004:**
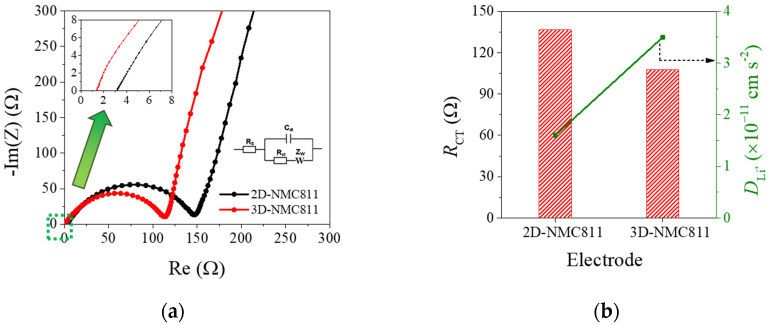
(**a**) Nyquist plots of 2D- and 3D-NMC811 electrodes before galvanostatic measurements; (**b**) Fitting results for impedance spectra and calculated *D*_Li+_ from EIS.

## Data Availability

Not applicable.

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
