# Peer review of "Ultrafast-Laser Micro-Structuring of LiNi0.8Mn0.1Co0.1O2 Cathode for High-Rate Capability of Three-Dimensional Li-ion Batteries"

_nanomaterials, 2022, doi:10.3390/nano12213897_

Round 1

Reviewer 1 Report

This paper compares the electrochemical performances of two electrodes for LI ion batteries, one is laser patterned (3D) and another is the original one (2D sample). The technique is not new in the field, but the results are indeed new and interesting. The results proved the effectiveness of laser surface treatment on improving the Li+ ion diffusion dynamics in LIBs due to the formation of 3D patterns, which could expose more surface area of electrode materials, compared to that of 2D electrode. I would recommend the acceptance of the manuscript for publication. However, I have few comments about the work:

1.   Using the laser beam to engrave the surface of electrode, you create those 3D channels to enhance the Li+ diffusion, the edges of those engraved channels can be very sharp, you may expect the concentration of E-field, and that may lead to the strong electromigration of Li+ ions toward those shape edges, then the Li dendrite could form, please comment in your paper about such mechanism.

2. Figure. 3, were there two different sets of equipment for I-V measurements?  (c) and (d) obviously give very smooth profiles, and (a) and (b) on the other hand, look not as beautiful as (c) and (d)?

3. In Page. 7,  you mentioned "the diffusivity coefficients of 2D-NMC cathode were 16.6 x 10^-11 and 16.6 x 10^-11 cm^2S^-1 for .....", can you check the numbers, are they indeed completely identical?

Author Response

Reviewer 1: This paper compares the electrochemical performances of two electrodes for LI ion batteries, one is laser patterned (3D) and another is the original one (2D sample). The technique is not new in the field, but the results are indeed new and interesting. The results proved the effectiveness of laser surface treatment on improving the Li+ ion diffusion dynamics in LIBs due to the formation of 3D patterns, which could expose more surface area of electrode materials, compared to that of 2D electrode. I would recommend the acceptance of the manuscript for publication. However, I have few comments about the work:

  1. Using the laser beam to engrave the surface of electrode, you create those 3D channels to enhance the Li+ diffusion, the edges of those engraved channels can be very sharp, you may expect the concentration of E-field, and that may lead to the strong electromigration of Li+ ions toward those shape edges, then the Li dendrite could form, please comment in your paper about such mechanism.

Answer: Thank you for your suggestion. The ultrafast laser beam ablates the electrode materials to generate channels with sharp edges as shown in the cross-sectional optical microscopic images (Figure 1c and d). This leads to the concentration of E-field at those edges, as the reviewer pointed out. However, unlike Li metal anode in which the uneven electric field leads to dendrite growth, this phenomena doesn’t affect significantly to NMC811 cathode, as it was operated within the high potential window of 3.0 – 4.3 V vs. Li+/Li. On the other hand, the laser structuring of cathode in this work can enhance Li+ ion transport across the thick electrode to provide fast pathways for facile Li+ ion transport and to limit electrolyte concentration gradients due to increased interfacial area.   

  1. Figure. 3, were there two different sets of equipment for I-V measurements?  (c) and (d) obviously give very smooth profiles, and (a) and (b) on the other hand, look not as beautiful as (c) and (d)?

Answer: Thank you for your suggestion. We used the same equipment for I-V measurements, which was shown in Figure 3. The sharp profiles as shown in Figure 3c and d came from higher scan rates.

  1. In Page. 7, you mentioned "the diffusivity coefficients of 2D-NMC cathode were 16.6 x 10^-11 and 16.6 x 10^-11 cm^2S^-1 for .....", can you check the numbers, are they indeed completely identical?

Answer: Thank you for your suggestion. The diffusivity coefficient for 2D-NMC cathode was and 2.0 × 10−11 cm2 s−1 for the reduction processes. The diffusivity coefficient value was corrected in the revised manuscript at Page 8.

Reviewer 2 Report

This article presents a study of preparation of 3D of Ni-rich LiNiMnCo-based cathode material prepared by femtosecond ultrafast-laser microstructure patterning. Commercially available cathode  was additionally modified to generate 3D-electrode configuration. Improved capacity was observed for thick-film material, which was not the case for thin-film.  It was presented how 3D-concept could be positive approach to increase rate capability and power output in batteries.

The idea seems interesting. I believe the topic might be interesting to readers. The language and writing style need to be work on. In the light of scientific content, there are some points that I feel should be addressed and clarified:

1.   Literature: Please, add up-to-date references where possible, especially in the Introduction part.

3.   Materials and Methods section: Please add more information and details. Regarding impedance spectroscopy, what about temperature tests? Why go to such low frequencies, mHz?

4.    Results: Figure S1(d) In EDS why there is maxima coming from Pt? Please explain

5.     Results: Share more details about 3D cathode template. Please explain

6.   Results: Did you think in employing NMR studies? for the diffusion coefficients? It would be interesting to see. 

7.  Please, increase the Figure quality, make them uniform in style and condense Figures where possible.

8.      I also suggest editing of English language and style.

Author Response

Reviewer 2: This article presents a study of preparation of 3D of Ni-rich LiNiMnCo-based cathode material prepared by femtosecond ultrafast-laser microstructure patterning. Commercially available cathode was additionally modified to generate 3D-electrode configuration. Improved capacity was observed for thick-film material, which was not the case for thin-film.  It was presented how 3D-concept could be positive approach to increase rate capability and power output in batteries.

The idea seems interesting. I believe the topic might be interesting to readers. The language and writing style need to be work on. In the light of scientific content, there are some points that I feel should be addressed and clarified:

  1. Literature: Please, add up-to-date references where possible, especially in the Introduction part.

Answer: Thank you for your suggestion. Updated references from 1 to 10 in references added to our revised manuscript in the introduction section.

  1. Materials and Methods section: Please add more information and details. Regarding impedance spectroscopy, what about temperature tests? Why go to such low frequencies, mHz?

Answer: Thank you for your suggestion. The impedance measurement was conducted at room temperature at 25 oC, but we didn’t investigate effect of temperature in our experimental scope. For impedance spectroscopy, the measurement was performed from low frequency of 1 mHz to high frequency of 100 kHz, in order to obtain the diffusion region (the slope of the Nyquist plots), which was used to calculate the diffusion coefficient, using the equation (2). The semi-circle radius of Nyquist plots corresponding to the interfacial resistance of the electrodes.

  1. Results: Figure S1(d) In EDS why there is maxima coming from Pt? Please explain

Answer: Thank you for your suggestion. The signal of Pt comes from the Pt sputter coating on the top surface of the sample, so as to enhance the signal-to-noise ratio during SEM imaging analysis experiment.

  1. Results: Share more details about 3D cathode template. Please explain

Answer: Thank you for your suggestion. The detailed description on 3D cathode templates in term of dimension and shape were added to the revised manuscript at Page 5 and 6. “The 40 µm-thick NMC811 channel dimensions were determined to be ~25 μm wide and ~38 μm deep, while the 100 µm-thick cathode channels were found to be ~25 μm wide and ~90 μm deep. The edges of the cathode channels were observed to be smooth and uniform.”

  1. Results: Did you think in employing NMR studies? for the diffusion coefficients? It would be interesting to see.

Answer: Thank you for your suggestion. The comparison of Li+ ion diffusion for both 2D- and 3D-electrodes was studied by two processes, by cyclic voltammetry and by electrochemical impedance spectroscopy. The results were consistent for both methods, whereas the laser structuring of electrode enhance the Li+ ion diffusion. Thus, although NMR studies (especially solid state NMR) may be helpful, but we could explain the differences in lithium diffusivity by using the aforementioned analysis systems.

  1. Please, increase the Figure quality, make them uniform in style and condense Figures where possible.

Answer: Thank you for your suggestion. All the figures were revised carefully and upgraded in our revised manuscript.

  1. I also suggest editing of English language and style.

Answer: Thank you for your suggestion. To meet your kind reminding, the English writing of this manuscript has been modified again carefully by a professional English language editing and publication support company, as below:

Reviewer 3 Report

Comments: In general, this work is of great significance for the further development of lithium ion battery cathode materials. However, there are still some questions that need to be answered before they can be accepted for publication.

1. The traditional binary oxide may have some problems, but what about the ternary oxide? The object of your research can be said to be quaternary oxide. Is the performance of five or more elemental oxides better?

2. Is the magnification of (a) and (b) in Figure 1 the same? I suggest that the author should show pictures at the same magnification, which is more meaningful for comparison.

3. Can you have more cycles in Figure 2?

4. Please supplement the relevant charge discharge curve.

Author Response

Reviewer 3: In general, this work is of great significance for the further development of lithium ion battery cathode materials. However, there are still some questions that need to be answered before they can be accepted for publication.

  1. The traditional binary oxide may have some problems, but what about the ternary oxide? The object of your research can be said to be quaternary oxide. Is the performance of five or more elemental oxides better?

Answer: Thank you for your suggestion. As ultrafast-laser micro-structuring is a versatile technology to be compatible with several cathodes (LiFePO4, LiMn2O4, LiNixMnyCozO2, etc.) and anodes (graphite, Sn, Si, etc.), it is probably applicable to quaternary metal oxide cathodes. The performances of structuring quaternary metal oxide cathodes may be varied according to the intrinsic differences of those materials to NMC. It can be concluded that the effect of the cathode materials (e.g. metal oxides) on electrochemical performances was insignificant in our experimental scope.

  1. Is the magnification of (a) and (b) in Figure 1 the same? I suggest that the author should show pictures at the same magnification, which is more meaningful for comparison.

Answer: Thank you for your suggestion. Figure 1b is the higher magnified of Figure 1a in order to give a clearer view of the structuring electrode in term of dimension and shape. So we explained in the magnification range in the figure caption.

  1. Can you have more cycles in Figure 2?

Answer: Thank you for your suggestion. The goal of this study is to investigate the effect of femtosecond ultra-fast laser structuring on the rate capability performances of NMC811 cathode materials for three-dimensional batteries. Thus, it was focused on rate capabilities at high current density and used electrochemical analysis to prove the enhancement in Li+ ion diffusion.

  1. Please supplement the relevant charge discharge curve.

Answer: Thank you for your suggestion. The relevant galvanostatic charge/ discharge curves for 2D-NMC811 and 3D-NMC811 prepared with 40-µm and 100-µm thickness, respectively, were added to the revised supporting information as Figure S3.

Round 2

Reviewer 3 Report

The author has improved and revised the article according to the requirements of the reviewer, and it is recommended to accept the manuscript.